# Inelastic electron scattering induced quantum coherence in molecular dynamics

Akshay Kumar[1], Suvasis Swain[1,2] & Vaibhav S. Prabhudesai [1] ✉

Quantum coherence is pivotal in various applications ranging from chemical control to quantum computing. An example of its manifestation in molecular dynamics is inversion symmetry breaking in the photodissociation of homonuclear diatomic molecules. On the other hand, the dissociative attachment of an incoherent electron also induces such coherent dynamics. However, these processes are resonant and occur for projectiles with a specific energy. Here we present the most general scenario of non-resonant inelastic electron scattering inducing such a quantum coherence in molecular dynamics. The ion-pair formation ($H^+ + H^-$) that proceeds after the electron impact excitation of $H_2$ shows a forward-backward asymmetry about the incoming electron beam. Simultaneous transfer of multiple angular momentum quanta during the electron collision induces the underlying coherence in the system. The non-resonant nature of this process makes this effect generic and points to its possible prevalent role in particle collision processes, including electron-induced chemistry.

In modern scientific endeavours, controlling chemical reactions has been one of the most sought-after goals. In this context, coherent control of molecular dynamics using light has been explored quite extensively since the invention of lasers. The coherence induced due to the absorption of photons forms the basis of the mechanism behind this control[1]. In this context, single electron attachment to a molecule also induces coherent molecular dynamics[2]. Here, the attachment of a free electron causes the transfer of multiple quanta of angular momentum to the molecule. These momentum transfer channels that originate due to the single electron attachment are inherently coherent, introducing coherence in the dissociation dynamics in the produced anion. Although this effect is induced in every such attachment process, its undisputed signature is observed in the homonuclear diatomic molecules. Such multiple momentum transfer channels that individually have inversion symmetry may invoke the formation of coherent superposition of such multiple anion resonance states. As these individual anion states are associated with transferring either an even or odd number of angular momentum quanta to the ground molecular state, they show inversion symmetric or antisymmetric characteristics. However, their coherent superposition would not possess the inversion symmetry. Dissociation resulting from such a superposition state shows inversion symmetry breaking in the form of forward-backward asymmetry in the fragment anion ejection with respect to the incoming electron beam. This process is analogous to the interference of two quantum paths invoked by the absorption of one or two photons. However, both photoabsorption and electron attachment are resonant processes. They occur at a specific photon or electron energy at which such respective excited neutral or anion states are present.

A common feature of free electron molecule collision is non-resonant inelastic scattering, in which the incoming electron excites the target molecule by transferring part of its kinetic energy. Inelastic electron collision with molecules takes place wherever free electrons are available. The most extensively studied phenomenon arising from inelastic electron scattering is electron impact ionisation. Electron attachment is also a subset of inelastic electron scattering. However, electron impact excitation and dissociation are equally important but relatively less explored channels. Except for the electron attachment, all other channels are non-resonant processes that open up if the incoming electron has more energy than their respective energy thresholds. For example, the ionisation potential for the $H_2$ molecule is 15.4 eV, and the inelastic scattering of an electron with higher energy

[1]Tata Institute of Fundamental Research, Colaba Mumbai 400005, India. [2]Present address: Centre de Recherche sur les Ions, les Matériaux et la Photonique (CIMAP) - UMR 6252 - Normandie Université, ENSICAEN, UNICAEN, CEA, CNRS, 14000 Caen, France. ✉e-mail: vaibhav@tifr.res.in

may cause its ionisation. These inelastic scattering channels are significant in plasma processes, radiation biology, astrochemistry, and any other environments involving energetic electrons capable of triggering these processes. They are responsible for the chemistry of such environments. More importantly, each such collision would associate with angular momentum transfer to the molecule, similar to what is observed in the electron attachment process. Then would they also invoke the coherent dynamics in the molecule?

Ion-pair formation, also known as dipolar dissociation, is a charge-asymmetric dissociation of the excited molecule. It is one of the channels of dissociation arising from the non-resonant inelastic scattering of particles. In this work, we show the inversion symmetry breaking in the ion-pair formation from molecular hydrogen on inelastic collision of electrons. We explain this result as due to the coherent excitation of two states of opposite parties that results in symmetry breaking. Ion-pair formation being a nonresonant inelastic process, the coherent transition to multiple excited states and the subsequent molecular dynamics happens at all electron energies beyond the threshold. Most importantly, as the electronic excitation of molecules is a very general process in all energetic particle-molecule collisions, these observations apply to all such cases.

## Results

We investigated the ion-pair formation in molecular hydrogen on electron collision. In $H_2$, this process would result in $H^+ + H^-$ formation. The minimum energy required for obtaining this channel is about 17.3 eV[3]. This process is understood as a predissociation of the excited $H_2$ molecule via the ion-pair potential energy curve of the system, as shown in Fig. 1(a) and (b). Hence, inelastic scattering of free electrons with an energy >17.3 eV would cause this process. The incoming electrons with sufficient energy excite the hydrogen molecule to the neutral states present above this threshold energy in the Franck-Condon region. On survival against autoionisation, these states may predissociate on the ion-pair curve to the ion-pair formation limit.

In principle, the ion-pair formation can be studied by observing either the $H^-$ or $H^+$ signal. However, from 18 eV onwards, $H^+$ is also produced by dissociative ionisation, which prevents obtaining a clean signal from the ion-pair formation channel[4,5]. Hence observing $H^-$ is the ideal way to study this process. The ion yield curve of the $H^-$ ions from $H_2$ shows a steady increase in the ion signal from 17.3 eV as this is a non-resonant process[6].

We obtained the velocity slice image (VSI) of $H^-$ ions as a function of electron energy starting from the threshold of the ion-pair formation. The experimental details are described in Methods. Starting from the 17.3 eV electron energy, the image shows a blob. As the electron energy increases, the blob size does not change. As the electron energy reaches 25 eV, a new structure in the form of a ring begins to appear in the VSI image, indicating the formation of ions with higher kinetic energy. This ring widens in its outer radius until the electron energy is 40 eV, and its size remains unchanged beyond it (see the Suplementary Information). We explain the appearance of these two structures and their behaviour based on the potential energy curves given in Fig. 1.

Energetically, the ion-pair formation limit lies above the neutral $H_2$ dissociation limit of $H(1s) + H(n = 2, 3, \text{ and } 4)$ (Fig. 1). Hence, only those curves dissociating to these limits would cross the ion-pair potential energy curve. Moreover, due to angular momentum consideration, the ion-pair formation limit $(H^+ + H^-(^1S))$ corresponds to only $^1\Sigma_u^+$ and $^1\Sigma_g^+$ states. Therefore, the excited target states of only these symmetries, dissociating to the $H(1s) + H(n = 2, 3, \text{ and } 4)$ limit, would contribute to the ion-pair formation. The series of states of these allowed symmetries converging to the ground state of the $H_2^+$ cation intersect the Franck-Condon region close to the threshold of ion-pair formation, as shown in Fig. 1a. On surviving the autoionisation to the ground cation state, these states contribute to the ion-pair formation at and near the threshold. The $H^-$ from these states will appear with very little kinetic energy. The kinetic energy of the $H^-$ ions produced would not change with the electron energy due to its origin from a threshold process, as the scattered electron would carry away the excess energy after the molecular excitation. The relatively smaller slope of the potential energy curves and their limited range in the Franck-Condon region also contribute to this behaviour, as shown in Fig. 1a.

The ring formed by ions of higher kinetic energy may be explained by considering the potential energy curves in Fig. 1b, where we show the set of higher-lying states that belong to the Rydberg series ($Q_1$ series)[7,8] converging to the first excited state of $H_2^+$. These states intersect the Franck-Condon region from 25 eV onwards. For these states as well, the dissociation limits are the same ($H(1s) + H(n = 2, 3, \text{ and } 4)$), and hence based on symmetry, only $^1\Sigma_g^+$ and $^1\Sigma_u^+$ states of the $Q_1$ series would contribute. As these states are considerably higher in energy than the dissociation limit, the ion-pair formed would carry substantial kinetic energy. These states also have a steep slope in the Franck-Condon region; consequently, the kinetic energy distribution would widen as a function of electron energy. However, beyond 40 eV,

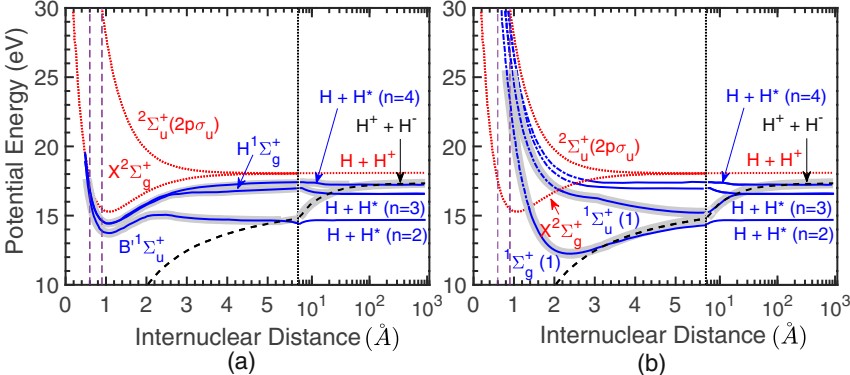

**Fig. 1 | Schematic of ion-pair formation in $H_2$.** The potential energy curves are taken from ref. 7. **a** For the production of low kinetic energy $H^-$ ions and (**b**) for the high kinetic energy ions. The potential energy scale is about the v = 0 state of the electronic ground state of $H_2$. The red dotted curves are the $H_2^+$ ground and first excited states. The black dashed curve is the ion-pair curve, and the blue curves are the neutral excited states from the Rydberg series converging to the cation ground state in (**a**) contributing to the low kinetic energy ion-pair signal and from the $Q_1$ series in (**b**) contributing to the higher kinetic energy ion-pair signal. The blue

curves from the $Q_1$ series are partly shown as dashed-dot, indicating that the autoionisation is active in this region. The grey thick-shaded curves indicate the paths taken by the $^1\Sigma_g^+$ and $^1\Sigma_u^+$ states after excitation by the incoming electrons leading to the ion-pair formation (see the text). Vertical purple dashed lines indicate the Franck-Condon region with the vibrational ground level of the electronic ground state of $H_2$. The verticle black dotted line separates the linear $x$-axis scale from the logarithmic $x$-axis scale for larger internuclear separation.

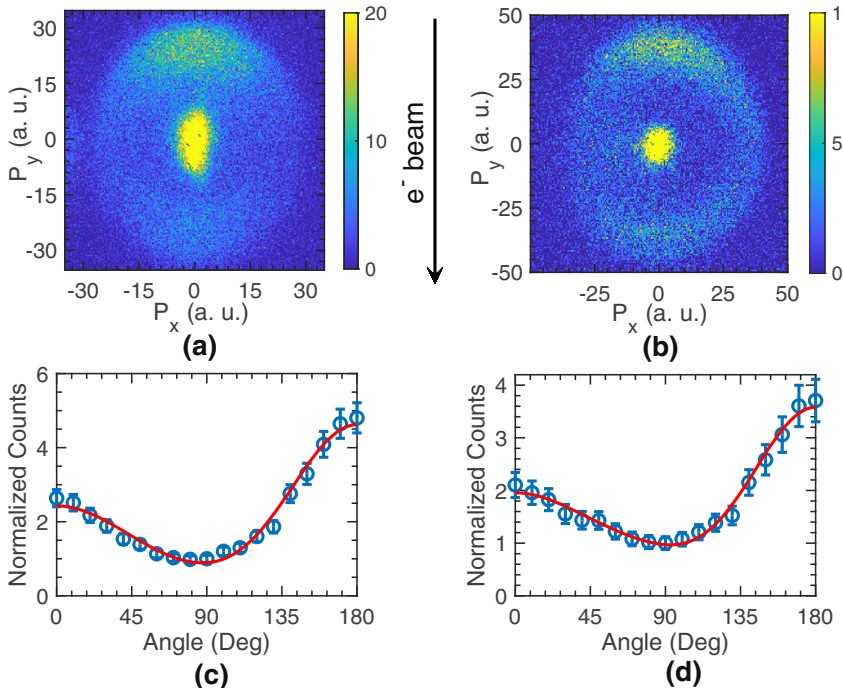

**(a)** **(b)**

**(c)** **(d)**

**Fig. 2 | Angular distribution of ion-pair formation.** Velocity slice image of (**a**) $H^-$ from $H_2$ and (**b**) $D^-$ from $D_2$ obtained from the ion-pair formation at the 50 eV electron energy. The arrow indicates the direction of the incoming electron beam. **c** and **d** Are the angular distributions obtained for the kinetic energy range of 2.5–9 eV from the two images (**a**) and (**b**), respectively. The angular distributions are normalised w.r.t. the counts at 90° with the error bars indicating the standard deviation. The solid (red) curves in (**c**) and (**d**) are the fits for the data obtained using Eq. 6, discussed in the text below. The error bars in (**c**) and (**d**) are determined by the error propagation of the counting errors for the negative ions from the momentum images in (**a**) and (**b**), respectively.

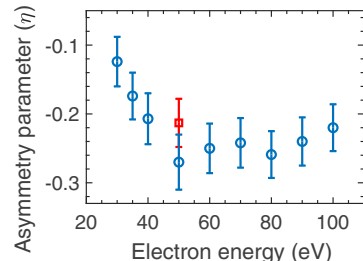

**Fig. 3 | Asymmetry parameter.** Experimentally obtained kinetic energy-integrated asymmetry parameter ($\eta$) for ion-pair formation from $H_2$ as a function of electron energy (blue circles). Also shown is the asymmetry parameter obtained for $D_2$ at 50 eV electron energy (red square). The error bars are obtained by the error pro-pagation of the counting errors of the ions measured in the forward and backward directions. The images used for determining these asymmetry parameters are given in the Supplementary Information.

these states would go out of the Franck-Condon region, explaining the observed behaviour of the ring with electron energy change.

Figure 2(a) and (b) show the VSIs obtained for $H_2$ and $D_2$ at 50 eV electron energy, respectively. Since the outer ring shows anisotropy in the angular distribution, we plotted them for the two isotopomers in Fig. 2(c) and (d), respectively. Here we have taken the intensity from the entire width of the ring (corresponding to a kinetic spread of 2.5 eV–9 eV). The angular distribution is obtained with respect to the incoming electron beam direction. It shows maxima in the forward and backward directions with a non-zero signal at all angles. Most importantly, the angular distribution shows forward-backward asymmetry with more intensity in the backward direction. We have taken care that this asymmetry is not arising from any experimental artefacts due to the recoil motion of the molecular target and detector efficiency (see

Methods). The asymmetry observed here is intriguing as it is not expected from a homonuclear diatomic molecule due to its inherent inversion symmetry, which must show up as a forward-backward symmetric dissociation pattern.

The overall signal strength is poorer in $D_2$ than in $H_2$ due to the isotope effect, where the heavier isotope experiences more loss in the population due to autoionisation, as pointed out by Krishnakumar et al.[7]. We have measured the forward-backward asymmetry in terms of the asymmetry parameter $\eta$ defined as

$$\eta = \frac{I_F - I_B}{I_F + I_B} \qquad (1)$$

where $I_F$ and $I_B$ are the signal intensities in the forward and backward direction with respect to the electron beam integrated over the kinetic energy range of 2.5–9 eV. Figure 3 shows the asymmetry parameter, $\eta$, obtained experimentally as a function of the incoming electron energy. It also shows the value of $\eta$ obtained for $D_2$ at 50 eV electron energy. We note that the asymmetry parameter changes with electron energy. It is also different for $H_2$ and $D_2$, though the measured uncertainties overlap.

The angular distribution of the molecular dissociation products following electron impact is discussed in detail by Van Brunt[9]. The angular distribution at the threshold, where the entire electron energy is transferred to the excitation of the molecule, is expected to follow that of the dissociative attachment (DA) process. However, as the electron energy increases and the scattered electron carries away the excess energy, the angular distribution is expected to show changes with the electron energy, predominantly due to the role of higher partial waves in the excitation process. As the initial and the final target states have specific inversion symmetry, the dissociation process is expected to retain this overall inversion symmetry. This implies that only odd or even partial waves play a role in the excitation process for a

homonuclear diatomic molecule depending on the initial and final state of the target. For example, in the case of $H_2$, the $^1\Sigma_g^+ \rightarrow {}^1\Sigma_g^+$ transition will have contributions only from the even partial waves, and that for the $^1\Sigma_g^+ \rightarrow {}^1\Sigma_u^+$ transition would be only odd partial waves. As a result, the transition probability would be an even function of the angle, and the angular distribution would always be symmetric about 90° to the incoming electron beam for any given dissociating excited state. Van Brunt provided the mathematical formulation to estimate the angular distribution of the ion-pair formation process[9]. For a single excited state contributing to the ion-pair formation at the threshold energy, the angular distribution of the fragment ion is given by

$$I(\theta, K) = K^{-n}\left| \sum_{l=\lceil \mu \rceil}^{\infty} i^l i^{m(l+1)} \sqrt{\frac{(2l+1)(l-\mu)!}{(l+\mu)!}} j_l(Kr) Y_{l,\mu}(\theta, \phi) \right|^2 \quad (2)$$

where $n$, $m$, and $r$ are adjustable parameters, $K$ is the magnitude of momentum transfer from electron to molecule, $j_l(Kr)$ is the spherical Bessel function, and $Y_{l,\mu}(\theta, \phi)$ are the spherical harmonics. Here, parameter $n$ is related to the partial wave involved in the transition, $m$ is related to the relative phase of the partial waves involved, and $r$ is related to the typical interaction length, which typically does not exceed 10 a.u.[9]. For the homonuclear diatomic molecule, $l$ values would be either odd or even depending on the inversion symmetry for the initial and final states. For the non-threshold values of the incoming electron energies, the effective angular distribution can be estimated by integrating the $I(\theta, K)$ as

$$I(\theta) = \int_{K_0 - K_f}^{K_0 + K_f} I(\theta, K) K dK \quad (3)$$

where $K_O$ is the wavenumber of the incoming electron and $K_f$ is the final wavenumber of the scattered electron.

If multiple states are contributing to the ion-pair formation incoherently at a given electron energy, the final angular distribution can be obtained as the sum of the individual contribution of each of these states given as

$$I(\theta) = \int_{K_0 - K_f}^{K_0 + K_f} \sum_j c_j I_j(\theta, K) K dK \quad (4)$$

where $I_j(\theta, K)$ corresponds to the angle-dependent contribution of the $j$th excited state obtained using Eq. (2), and $c_j$ is the corresponding weightage.

Based on Eqs. (3) and (4), the observed forward-backward asymmetry in the angular distribution cannot be explained using just one of the $^1\Sigma_g^+$ or $^1\Sigma_u^+$ states or by adding their contributions incoherently. This is because each of the terms in Eq. (4) will be symmetric under inversion. In the following, we explain the observed asymmetry by assuming a coherent excitation process in which a superposition of two states of opposite parities is created. This necessitates the use of both even and odd partial wave contributions coherently in the transition probability. That is, the angular distribution observed is needed to be expressed as the square of the sum of the amplitudes from the individual state rather than the sum of the square of the amplitudes. This implies that the angular distribution may be written as

$$I(\theta) = \int_{K_0 - K_f}^{K_0 + K_f} K^{-n}\left| \sum_j a_j \lambda_j(\theta, K) \right|^2 K dK \quad (5)$$

Here, $\lambda_j$ is the amplitude corresponding to the contribution from the $j$th excited state according to Eq. (2), and $a_j$ is the complex coefficient that carries the information of the weightage of the state and the phase w.r.t. the other contributing states.

We have taken $n = 6$ as suggested in ref. [9], although the final function does not vary much with this. For the $^1\Sigma_g^+ \rightarrow {}^1\Sigma_u^+$ transition, we have taken m = 1, and for the $^1\Sigma_g^+ \rightarrow {}^1\Sigma_g^+$ transition, we take m = 0[9]. The lower partial waves play a significant role close to the threshold, i.e. electron energy close to the excitation energy of the molecular state. However, as the electron energy increases, higher partial waves may become dominant with a higher value of $Kr$. As the observed angular distribution does not show any sharp structures, we restrict our angular distribution functions to the lower allowed partial waves for each transition. Hence, after considering the $s$ and $d$-wave contribution in the $^1\Sigma_g^+ \rightarrow {}^1\Sigma_g^+$ transition and $p$-wave contribution to the $^1\Sigma_g^+ \rightarrow {}^1\Sigma_u^+$ transition, we obtain the fitting function for the observed angular distribution using Eq. (5) as

$$\begin{aligned} I(\theta) = \int_{K_0 - K_f}^{K_0 + K_f} K^{-6} | a(j_0(Kr) Y_{00}(\theta, \varphi) + j_2(Kr)\sqrt{5} Y_{2,0}(\theta, \varphi) e^{-i\phi_2}) \\ + b\sqrt{3} j_1(Kr) Y_{1,0}(\theta, \varphi) e^{-i\phi_1} |^2 K dK \end{aligned} \quad (6)$$

Here, $\phi_1$ is the relative phase between the $^1\Sigma_g^+ \rightarrow {}^1\Sigma_g^+$ and $^1\Sigma_g^+ \rightarrow {}^1\Sigma_u^+$ transitions, and $\phi_2$ is the relative phase between the $s$-wave and $d$-wave contribution in the $^1\Sigma_g^+ \rightarrow {}^1\Sigma_g^+$ transition. This includes the intrinsic phase difference between the $s$-wave and $p$-wave from the same electron and the relative phase difference gained during the dissociation process along the two dissociation paths. The obtained fit for the data using Eq. (6) is shown in Fig. 2(c) for $H_2$ and 2(d) for $D_2$ at 50 eV electron energy and $r = 1.5$ a.u.

## Model for estimating the forward-backward asymmetry

As mentioned above, the states that produce the faster $H^-$ ions by the ion-pair formation process belong to the $Q_I$ series of autoionising states, which are repulsive in the Franck-Condon region[8]. Due to predissociation on the ion-pair potential energy curve, these states contribute to the measured $H^-$ channel. As explained earlier, the Franck-Condon overlap with these states extends from 25 eV to 40 eV, yielding dissociating products with the kinetic energies observed in the outer ring of the VSI obtained in our experiment. Guberman calculated the potential energy curves for these states[10]. Sanchez and Martin have also calculated these curves and their width towards autoionisation[11]. We have used these widths and the composite potential energy curve interpolated from the curves compiled by Vogel[7].

The ion-pair formation process is analogous to the DA process as both processes compete with the decay of the underlying states by electron ejection. For DA, the parent negative ion resonance state may decay by autodetachment of the extra electron, whereas in ion-pair formation, the neutral excited state may decay by autoionisation. However, the most crucial difference between the two processes is that DA is a resonant process that involves a molecular negative ion state. In contrast, ion-pair formation is a non-resonant process and occurs at any electron energy above its threshold. In the following model, we consider only the lowest $^1\Sigma_g^+$ and $^1\Sigma_u^+$ states in the Franck-Condon region from the $Q_I$ series (Fig. 1(b)). We estimate the initial population of each of the states as a function of the transferred energy as being proportional to the square of the corresponding part of the ground state vibrational wavefunction in the Franck-Condon region. Using autoionization width as a function of internuclear separation from ref. [11], we determine the amplitude of each state that survives autoionisation. Subsequently, this wavepacket dissociates along the ion-pair curve due to curve crossing. We estimate the effective amplitude of the dissociating wavepackets along the ion-pair curve by multiplying the surviving wavepacket amplitude by the Landau-Zener factor that gives the transition

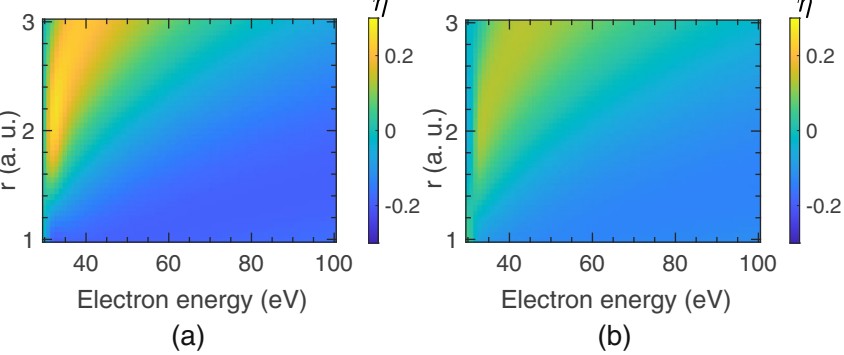

**Fig. 4 | Asymmetry parameter (η) from the model.** The asymmetry parameter ($\eta$) as false colour plots obtained from the model for (**a**) $H_2$ and (**b**) $D_2$ as a function of incoming electron energy for various values of the '$r$' parameter.

probability between the diabatic states as[12]

$$P_{ij} = 1 - \exp\left(\frac{-2\pi c_{ij}^2}{\hbar\alpha v}\right) \qquad (7)$$

where $c_{ij}$ is the coupling matrix element between the crossing states $i$ and $j$. This coupling matrix element effectively becomes half of the closest energy separation between the two relevant adiabatic potential energy curves. The typical half-splitting value for $H_2$ is about 0.27 eV[13]. The absolute slope difference (at the crossing) between the adiabatic potentials is denoted as $\alpha$, and $v$ is the relative velocity of the fragments. We estimate the phase difference between the two paths along the two $Q_1$ potential energy curves that cross the ion-pair curve at higher internuclear distances (grey-shaded curves in Fig. 1(b)). For both paths, we have used the same ion-pair curve. Using this phase difference and the remaining amplitudes of the two channels, we estimate the expected forward-backward asymmetry from the angular distribution function

$$I(\theta) = \int_{K_0-K_f}^{K_0+K_f} K^{-6} |a_g(j_0(Kr)Y_{00}(\theta,\varphi) + j_2(Kr)\sqrt{5}Y_{2,0}(\theta,\varphi)e^{-i\phi_2})$$
$$+ a_u\sqrt{3}j_1(Kr)Y_{1,0}(\theta,\varphi)e^{-i\phi_1}|^2 K\,dK \qquad (8)$$

where the $a_g$ and $a_u$ are the effective amplitudes in each channel after the loss due to autoionisation and appropriate transition to the corresponding ion-pair states. We take $\phi_1$ as the sum of the relative phase gained during the dissociation along the two paths and the initial relative phase between the $s$ and $p$-waves, which is $\pi/2$, and $\phi_2$ as the initial phase difference between the $s$ and $d$-waves, which is $\pi$. The asymmetry parameter $\eta$ is determined according to Eq. (1). The $I_F$ and $I_B$ are obtained by integrating $I(\theta)$ from Eq. (8) over the range 0 to $\pi/2$ and $\pi/2$ to $\pi$, respectively, and summing over the values of $K_f$. Here, parameter r, equivalent to the impact parameter, does not have a well-defined value. We estimate the asymmetry parameter over a range of values for r. The estimated asymmetry parameter as a function of incident electron energy and the impact parameter range is shown in Fig. 4(a) for $H_2$.

We also show the estimated values for $D_2$, which shows the isotope effect (Fig. 4(b)). The heavier isotope yields reduced amplitudes of the interfering wavepackets and alter the relative phase between them. Due to steep dissociation, which results in high kinetic energy, we do not see a substantial difference between the two isotopes. However, the asymmetry diminishes for the heavier isotope as the width of the participating $^1\Sigma_g^+$ state is almost 50% larger than the $^1\Sigma_u^+$ state, affecting the contrast of the interference[11]. The values obtained by the model are lower than the measured asymmetry parameters (Fig. 3). However, the trend in the observed asymmetry parameters can easily

be observed in the simulated results for the typical impact parameters comparable to the equilibrium bond length of the molecule.

To conclude, we have shown that the non-resonant inelastic scattering of free electrons induces a coherent response from the molecule. This manifests in the ion-pair formation process in $H_2$ in the form of the inversion symmetry-breaking. The quantum coherence induced by the transfer of odd and even partial waves results in the transition of the molecule to the superposition of two states from the $Q_1$ band with opposite parity and dissociating to the same limit. The non-resonant nature of this process makes these results more generic than the DA or photodissociation processes that are resonant in nature. The above results indicate that similar asymmetry should be expected in the electron impact dissociation of inversion symmetric molecules resulting in the fragments in the ground and excited states, for example, in the electron impact dissociation of $H_2$ into $H + H^*$. We may point out that the forward-backward asymmetry observed in ion-pair formation from electron impact with $O_2$[14–16] may also be due to quantum interference of two dissociation paths, as seen in the present case of $H_2$, though it was attributed to molecular recoil effects[14] and detector efficiency problem[16]. Unlike $H_2$, $O_2$ is a far more complex system, making analysis of the data very difficult. However, the fact that such an asymmetry is not observed in the two-photon absorption process leading to the ion-pair formation in $O_2$ where multiple paths of opposite parities are *not* involved in the dissociation process[17] indicates the role of coherent excitation of states giving rise to the observed asymmetry $O_2$ data as well.

The most striking and essential aspect of these results is the conclusive identification of the coherent response of the matter to the inelastic scattering of the particles in terms of ensuing molecular dynamics. Although the coherence induced in the non-resonant inelastic scattering of electrons from an incoherent source is clearly observed in the homonuclear diatomic system, such effects must be prevalent in general in the electron scattering from any molecule. These findings also indicate that such coherent dynamics can be induced in the inelastic scattering of any particle with molecules. Recently, low-energy free-electron attachment has also been shown to control molecular dissociation, which is the first step towards realising the ultimate control in the chemical reaction[18]. In addition, there have been extensive explorations of low-energy electron-induced control over molecular dynamics, including efforts toward single-molecule engineering using a scanning tunnelling microscope[19]. Our results point to a possible avenue of electron-induced chemical control via the neutral excited states. However, it remains to be seen how we can tap these coherences to obtain total control over chemical reactions.

## Methods
### Experimental setup
The schematics of the experimental setup are shown in Fig. 5. The experiments were performed in a vacuum at a typical background

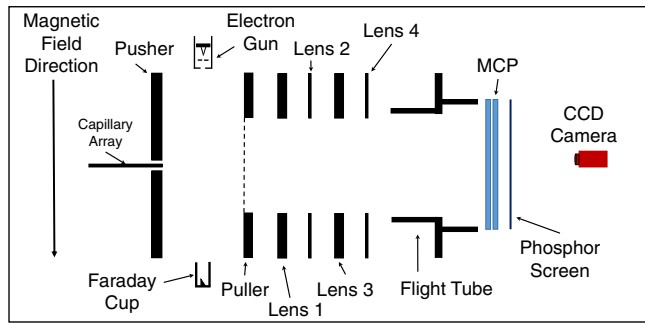

**Fig. 5 | Experimental setup.** Schematics of the experimental setup used. The capillary array produces the effusive molecular beam, which is crossed with the magnetically collimated pulsed electron beam produced by the home-built electron gun. The VSI spectrometer mounted coaxially with the molecular beam consists of four electrostatic lens-assembly followed by a short flight tube. The ions generated and extracted are detected by the position-sensitive detector comprised of a pair of micro-channel plates and a phosphor screen. The ion hits on the screen are recorded using a CCD camera.

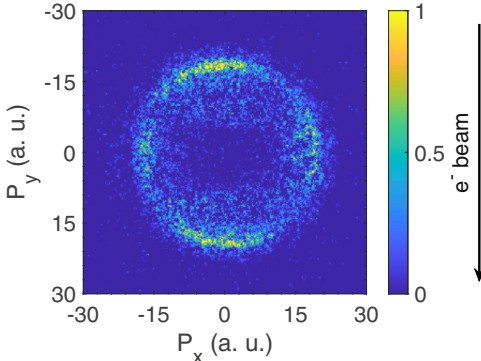

**Fig. 6 | Momentum image of DA at 8.5 eV.** Velocity slice image of $H^-$ ions obtained from the DA to $H_2$ at 8.5 eV electron energy. The direction of the electron beam is from top to bottom.

pressure of around $1 \times 10^{-6}$ torr after introducing the target gas. A pulsed (100 ns and 10 kHz repetition rate) electron beam collimated by a 50 G uniform magnetic field was made to cross an effusive molecular beam produced by a capillary array. The capillary array was mounted coaxially to the VSI spectrometer. The electron beam was created using a home-built electron gun that uses the tungsten filament. The electron gun current was measured using a Faraday cup mounted coaxially but diametrically opposite to the gun about the molecular beam. The magnetic field was generated using a pair of coils mounted in the Helmholtz geometry outside the vacuum chamber. We machined the entire vacuum chamber, along with the VSI spectrometer from titanium alloy of grade 'V', to avoid any effect of the stray magnetic field arising due to the magnetisation of the electrodes. The ions generated by electron interaction with molecules were extracted into the VSI spectrometer using a pulsed extraction field. This field was generated by applying a negative square voltage pulse (−350 V amplitude and 250 ns duration) on the pusher electrode, which was delayed from the electron pulse by 160 ns. The puller electrode was grounded, and appropriate DC voltages were applied to the lens electrodes along with the flight tube. The values of these voltages were first determined using an ion trajectory simulating software SIMION 8.0 and then optimised to obtain the proper image. As the electron beam collimating magnetic field was at right angles to the trajectories of the extracted ions, the trajectories deviated from the axis of the spectrometer. This deviation was maximum for the $H^-$ ions and could distort one side of the momentum image. We used a short-flight tube of length 15 mm to minimise this distortion. This short-flight tube was sufficient to separate the $H^-$ and $D^-$ ions in the time of flight spectrum obtained in the VSI mode. As the system carries the cylindrical symmetry about the electron beam axis, only the left or right half of the image that arrives close to the centre of the detector was used for the analysis.

We detected the ions using a two-dimensional position-sensitive detector made of a pair of microchannel plates followed by the phosphor screen. We pulsed the detector bias to obtain the slice of Newton's sphere of the ions generated in the interaction volume and extracted into the VSI spectrometer. The detector bias pulse was appropriately delayed w.r.t. the extraction pulse so that the central slice of the ion momentum sphere was captured on the detector. As the $H^-$ ions generated in the ion-pair formation process were relatively fast (KE up to 10 eV), we needed to use a higher extraction field along with the higher voltages on various electrodes suitable for the VSI. This reduced the overall flight time spread for the $H^-$ ions (to about 200 ns). To obtain meaningful slices from such a time of flight peak, we used a 10 ns duration detector biasing pulse. The images recorded using the

charge-coupled device camera were analysed offline for the $H^-$ ions' kinetic energy and angular distribution. We carried out the electron energy calibration using the DA signal from $H_2$ at 14 eV and the imaging calibration by measuring the VSI of $H^-$ ions from the 10 eV DA peak of $H_2$. The images are obtained from the crossed-beam geometry of the target region by subtracting the contribution from the static gas background[20].

We obtained the momentum image for the 10 eV DA resonance from $H_2$ that showed no asymmetry in the forward-backward direction. The image obtained at 8.5 eV is shown in Fig. 6. This observation is consistent with the fact that only one anion state contributes to the DA signal below 10 eV[20]. This eliminates any role of detector efficiency-induced artefact in the observed forward-backward asymmetry.

In the inelastic scattering process, the scattered electron induces the recoil motion in the molecular target, which influences the momentum distribution of the fragments in the laboratory frame. This recoil effect can appear as a small forward-backward asymmetry if the observation is made at a specific kinetic energy of the fragment[14,21]. The velocity slice imaging (VSI) technique provides the angular distribution for all the kinetic energies in a single measurement. We observe the forward-backward asymmetry for the signal integrated over the entire kinetic energy range, eliminating any such artefact due to molecular recoil.

## Data availability
The data that support the findings presented in Fig. 3 are provided in the Supplementary Information. The data are available from the corresponding author upon request.

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

## Acknowledgements

We thank E. Krishnakumar for valuable discussions. All authors acknowledge the financial support from the Dept. of Atomic Energy, India, under Project Identification No. RTI4002.

## Author contributions

V.S.P. planned the research. V.S.P. and S.S. built the experiment with help from A.K. A.K. and S.S. carried out the measurements. A.K. and V.S.P. carried out the simulations and analysis, and V.S.P. and A.K. interpreted the results. V.S.P. prepared the paper.

## Competing interests

The authors declare no competing interests.
