## [Peer Review File · Nature Communications]

Inelastic electron scattering induced quantum coherence in molecular dynamicsReviewer #1 (Remarks to the Author):

This work shows that the non-resonant inelastic collision of H₂ with an electron can lead to forward-backward symmetry breaking as the result of quantum coherent molecular dynamics following the electron impact. This experiment, which is claimed to be the first of its kind, has become possible due to improved resolution compared to a 2009 experiment (Ref. 6 of manuscript). The D₂ vs H₂ isotope effect is also observed.

The results are explained and quantified using a model from 1974 (Ref 9) and recent potential energy curves for H₂/H₂⁺ (Ref 5).

Even though there is already some theoretical understanding of this system, I believe the current experimental results are very valuable, as they quantitatively test the understanding for the theoretically perhaps more accessible system. The more so, since electron - molecule scattering is of great importance in many environments, as explained in the introductory part of the manuscript.

The manuscript also points out the "coherent chemical control" aspect of the work. Details of possible applications are not given, but I agree that the technical capability demonstrated in this work is valuable for the advancement of this field.

I am slightly confused by the text at the top of page 3 (lines 64-66): First it is mentioned that the ion-pair formation limit (H⁺ + H⁻) corresponds to $2\sigma^+_{u/g}$ (of the H + H⁺ system). That part is clear. The next sentence, however, mentions "excited target states of only these symmetries"... Here I get a little confused: "symmetries" seems to refer to $2\sigma^+_{u/g}$, which is the symmetry of the ionized states (H + H⁺), but it was my understanding that the excited states of the neutral H(1s)+H(n) are meant here?

When this issue is addressed, I recommend publication of this work in Nature Communications.

Reviewer #2 (Remarks to the Author):

See attached pdf file.

Reviewer #2 Attachment on the following page.

In their review of the first version of this manuscript, reviewer #2 also added some comments to the manuscript file. These comments were forwarded to the authors, who replied as included in this Peer Review File

Referee report: Inelastic electron scattering induced quantum coherence in molecular dynamics

A. Kumar, S. Swain, and V. S. Prabhudesai

The authors report on observations of forward-backward asymmetric angular scattering distributions in the H^- products of electron-induced ion-pair formation in H_2 and D_2 . This is attributed to interference between multiple coherently excited angular momentum states of the parent molecule. As the authors note towards the end of the manuscript, similar effects have been seen previously in molecular oxygen as long ago as 1974, when they were also attributed to interference between multiple partial waves, and also more recently in 2018 using the same velocity-map imaging technique used in the present manuscript.

This is a really nice piece of work, with clear experimental results backed up by theory and modelling. However, it is a judgement call whether the work is sufficiently novel for publication in *Nature Communications*, and in its present form I do not believe the authors make sufficiently strong arguments for this in light of previous work. If the article were to be published then I would suggest some significant rewriting and reorganisation of material in order to streamline the flow of information and make the manuscript more straightforward for the reader to follow. At present there is some jumping back and forth between the data, the interpretation, the theory, and the modelling, which tends to require the reader to jump around in the manuscript as well. I would suggest first presenting the data and working through the two observed dissociation channels and the proposed mechanisms for these, in terms of the relevant potential energy surfaces. I have to admit that I found the discussion of the potential energy surfaces involved rather difficult to follow. The potential energy curves shown in Figure 1 appear only to explain the higher kinetic energy product channel, and it is still not clear to me what the proposed mechanism is for formation of the low kinetic energy H^- ions that are also observed in the experiment. The authors could then continue on to explain the phenomenon that leads to the observed asymmetry in the angular distribution, and then describe the previously-developed theory and their own modelling. For the benefit of those not familiar with the various processes resulting from electron-molecule collisions and their relative cross-sections, in the Methods section it might also be worth briefly explaining why only the negative ion is detected, rather than the positive ion.

There are a number of other minor comments and suggested rewordings that would also help with clarity. I will address these by uploading an annotated version of the manuscript with my review rather than providing a long list here.

Response to the review reports

We thank the reviewers for their very constructive comments and suggestions. Our response to their comments/suggestions is given below.

Reviewer 1:

Comment:

This work shows that the non-resonant inelastic collision of H₂ with an electron can lead to forward-backward symmetry breaking as the result of quantum coherent molecular dynamics following the electron impact. This experiment, which is claimed to be the first of its kind, has become possible due to improved resolution compared to a 2009 experiment (Ref. 6 of manuscript). The D₂ vs H₂ isotope effect is also observed.

Response:

The experiment described in Ref. 6 of the original manuscript reports the asymmetry observed in the resonant dissociative photoionisation where the direct ionisation and autoionisation channels interfere. In our experiment, we show the symmetry breaking in the molecular dissociation dynamics due to the non-resonant inelastic scattering of a single electron. This non-resonant scattering leads to the excitation of the H₂ molecule to the coherent superposition of the two neutral excited states dissociating to the same limit. The very non-resonant nature of this process makes it more general and interesting from the electron scattering and induced chemistry point of view.

Comment:

The results are explained and quantified using a model from 1974 (Ref 9) and recent potential energy curves for H₂/H₂⁺ (Ref 5).

Even though there is already some theoretical understanding of this system, I believe the current experimental results are very valuable, as they quantitatively test the understanding for the theoretically perhaps more accessible system. The more so since electron-molecule scattering is of great importance in many environments, as explained in the introductory part of the manuscript.

The manuscript also points out the "coherent chemical control" aspect of the work. Details of possible applications are not given, but I agree that the technical capability demonstrated in this work is valuable for the advancement of this field.

Response:

We thank the reviewer for the encouraging and appreciative comments on the significance of the work presented in the manuscript.

Comment:

I am slightly confused by the text at the top of page 3 (lines 64-66): First, it is mentioned that the ion-pair formation limit (H⁺ + H⁻) corresponds to $2\sigma^+_{u/g}$ (of the H + H⁺ system). That part is clear. The next sentence, however, mentions "excited target states of only these symmetries"... Here, I get a little confused: "symmetries" seems to refer to " $2\sigma^+_{u/g}$ ", which is the symmetry of the ionised states (H

+ H+), but it was my understanding that the excited states of the neutral H(1s)+H(n) are meant here? When this issue is addressed, I recommend the publication of this work in Nature Communications.

Response:

We agree with the reviewer that there is an error in the molecular states described. The ion-pair states are of ${}^1\Sigma_{u/g}^+$ symmetry, and we have corrected the text in the manuscript on page 3. We thank the reviewer for pointing out this mistake and recommending our work for publication.

Reviewer 2:

Comment:

The authors report on observations of forward-backward asymmetric angular scattering distributions in the H products of electron-induced ion-pair formation in H₂ and D₂. This is attributed to interference between multiple coherently excited angular momentum states of the parent molecule. As the authors note towards the end of the manuscript, similar effects have been seen previously in molecular oxygen as long ago as 1974, when they were also attributed to interference between multiple partial waves, and also more recently in 2018, using the same velocity-map imaging technique used in the present manuscript.

Response:

We disagree with the assessment underlined in the comment. The 1974 paper (ref 14 in the original manuscript) reports the forward-backward asymmetry in the ion-pair formation from O₂. However, they measured the angular distribution in a narrow kinetic energy range and attributed this asymmetry to the momentum transfer from the electron (the recoil effect). The 2018 report (ref. 16 in the manuscript) used the velocity slice imaging technique to obtain the angular distribution and attributed the observed asymmetry to non-uniform detector efficiency. In both these reports, there was no mention of quantum coherence contributing to the asymmetry. However, this asymmetry can only be explained by invoking coherent excitation of multiple states. We feel that the reviewer's statement that those observations were attributed to interference between multiple partial waves is due to some confusion. Multiple partial waves may contribute even for a single excited state of the neutral molecule undergoing the ion pair formation, as given in Eq 2 of the manuscript. However, as we had already explained in our manuscript, for a homonuclear diatomic molecule, **either only even or only odd l values are allowed for a given excited state.** The corresponding angular distribution would always be an even function and hence will not show forward-backward asymmetry. Even considering two excited states, one of which may contribute even partial waves and the other odd partial waves, we cannot get the observed asymmetry unless they are created coherently, as explained in our manuscript. For additional clarity, we have explained this more explicitly in the revised manuscript by adding a couple of equations on page 7.

It is important to note that molecular oxygen is a far too complex system with many electrons and, consequently, very little information on the relevant excited states. In addition, the ion pair formation from it can lead to different states of O⁺. Therefore, modelling the process as we have done in the case of the two-electron system of H₂ is almost impossible. In the present experiment, we have systematically ruled out any instrumentation-based artefacts (see Methods). More importantly, H₂, the simplest molecule with only two electrons, allows us to use symmetry-based arguments to interpret the results and to identify the coherent process in the inelastic excitation process for the **first time.**

Comment:

This is a really nice piece of work, with clear experimental results backed up by theory and modelling. However, it is a judgement call whether the work is sufficiently novel for publication in Nature Communications, and in its present form, I do not believe the authors make sufficiently strong arguments for this in light of previous work.

Response:

We thank the reviewer for appreciating the experimental results presented in the manuscript. However, the reviewer is doubtful about the novelty of the present work in the continuation of his previous comments based on earlier work on O₂ that we have addressed above. To repeat, though the measurements on O₂ showed asymmetry, there is not even a hint of suggestion in the literature so far about coherent excitation of multiple states giving rise to it, neither by those authors nor anyone else. Only by making systematic measurements on H₂ and analysing the results as done by us could the role of coherent excitation be deduced.

Ion pair production is one of the inelastic processes which can be induced by the energetic collision of electrons, ions, or even neutrals with molecules. Though we have shown the existence of coherent excitation using electrons, it is applicable to the most general case of particle collisions. This, we believe, is a highly novel finding. There are three essential aspects to this work warranting publication in Nature Comm. (a) It is the first time that the idea of a coherent excitation process in the general category of non-resonant inelastic particle collision is being reported in the molecules; (b) even after more than 100 years of its history, fresh facets are being revealed in the general area of atomic collision phenomena as in the current report due to the recent developments in instrumentation; and (c) the inelastic collisions dominate the atomic collision processes starting from the formation of first-generation stars to varieties of applications in present-day technologies, where the coherence induced molecular dynamics has not been considered yet.

Comment:

If the article were to be published then I would suggest some significant rewriting and reorganisation of material in order to streamline the flow of information and make the manuscript more straightforward for the reader to follow. At present there is some jumping back and forth between the data, the interpretation, the theory, and the modelling, which tends to require the reader to jump around in the manuscript as well. I would suggest first presenting the data and working through the two observed dissociation channels and the proposed mechanisms for these, in terms of the relevant potential energy surfaces.

Response:

We thank the reviewer for this valuable suggestion of restructuring the manuscript. We agree with it and have now restructured the manuscript.

Comment:

I have to admit that I found the discussion of the potential energy surfaces involved rather difficult to follow. The potential energy curves shown in Figure 1 appear only to explain the higher kinetic energy product channel, and it is still not clear to me what the proposed mechanism is for formation of the low kinetic energy H⁻ ions that are also observed in the experiment. The authors could then continue on to

explain the phenomenon that leads to the observed asymmetry in the angular distribution, and then describe the previously-developed theory and their own modelling.

Response:

We agree with the reviewer about providing more clarity on the ion pair formation mechanisms. We have now revised the manuscript with an additional figure (figure 1(a)) to discuss the mechanism behind the formation of low-energy ions. The old figure 1 is now labelled as figure 1(b), and it is referred to while describing the higher energy ions. We have also revised the manuscript based on the other suggestions above.

Comment:

For the benefit of those not familiar with the various processes resulting from electron-molecule collisions and their relative cross-sections, in the Methods section it might also be worth briefly explaining why only the negative ion is detected, rather than the positive ion.

Response:

In the revised manuscript, we have added this explanation in the early part of the 'Results' section before discussing the results obtained for the H⁻ ions.

Comment:

There are a number of other minor comments and suggested rewordings that would also help with clarity. I will address these by uploading an annotated version of the manuscript with my review rather than providing a long list here.

Response:

We thank the reviewer for annotating the comments on the manuscript document, which we now address point by point. Most of the suggestions by the reviewer are adhered to in the manuscript revision. We also provide below the response to those comments that we did not agree with.

Comments annotated on the manuscript and the responses to them are listed below. We first address the major comments followed by the minor ones:

Line 14:

Worth mentioning the associated H⁺, which must have the same distribution by momentum conservation.

Response:

We agree with the reviewer that the H⁺ arising from the ion-pair formation will also show such asymmetry. However, as the H⁺ signal from the ion-pair formation would overlap with the H⁺ signal from the dissociative ionisation, it would be difficult to measure the asymmetry directly.

We have now changed the statement in the abstract:

'The ion-pair formation (H⁺ + H⁻) that proceeds after the electron impact excitation of H₂ shows a forward-backward asymmetry in the H⁻ ejection about the incoming electron beam.'

With:

The ion-pair formation ($H^+ + H^-$) that proceeds after the electron impact excitation of H_2 shows a forward-backward asymmetry about the incoming electron beam.

We have also added a statement related to the positive ions in the initial part of the 'Results' section in the revised manuscript.

Line 60:

'excites the hydrogen molecule to the state above': which state?

Response:

We have now modified the statement as follows:

*'excites the hydrogen molecule to the **neutral states present** above this threshold energy **in the Franck-Condon region.**'*

Line 65:

Surely the states mentioned should be singlet states for the ion pair.

Response:

The reviewer is correct that the involved states are singlet states; we have corrected the mistake.

Line 70 and 75:

The statements:

The kinetic energy of the H^- ions produced would not change with the electron energy due to its origin from a threshold process.

And

It is consistent with any non-resonant process where the scattered electron carries the remaining energy of the system.

Are not clear and need explanation.

Response:

We have added the explanations and restructured these paragraphs in the revised manuscript. We have added the following text to the relevant place.

The kinetic energy of the H^- ions produced would not change with the electron energy due to its origin from a threshold process, as the scattered electron would carry away the excess energy after the molecular excitation. The relatively smaller slope of the potential energy curves and their limited range in the Franck-Condon region also contribute to this behaviour, as shown in Fig. 1a.

Figure 1:

This diagram is for the higher E channel. Makes more sense. Which state is the molecule excited to at the 17.3 eV threshold? The grey pathways shown start at ~22 and 25 eV, respectively.

Response:

We agree with the reviewer that figure 1 does not explain the low kinetic energy path of the ion-pair formation. We have now added another figure (Fig 1(a) in the revised manuscript) to address this issue. We have also corrected Fig 1 (now Fig 1(b)) in the revised manuscript showing both paths starting from 25 eV.

Figure 2:

Response:

As suggested by the reviewer, we have added the 'e- beam' description on the arrow in the figure.

Line 121 and subsequent paragraph:

Most investigators use e2e spectroscopy, in which (we presume molecular recoil) can give rise to artefacts. Make it clear that this can be the same (kind) of artefact in other experiments (reported earlier), not this one.

Response:

We have now modified the manuscript with some rearrangements and have added the following text at the end of the 'Methods' section explaining this point:

In the inelastic scattering process, the scattered electron induces the recoil motion in the molecular target, which influences the momentum distribution of the fragments in the laboratory frame. This recoil effect can appear as a small forward-backward asymmetry if the observation is made at a specific kinetic energy of the fragment [14, 21]. The velocity slice imaging (VSI) technique provides the angular distribution for all the kinetic energies in a single measurement. We observe the forward-backward asymmetry for the signal integrated over the entire kinetic energy range, eliminating any such artefact due to molecular recoil.

Line 144:

The parameters 'n, m, and r' are related to what?

Response:

We have added the following text explaining the significance of these parameters.

Here, parameter n is related to the partial wave involved in the transition, m is related to the relative phase of the partial waves involved, and r is related to the typical interaction length, which typically does not exceed 10 au [9].

Lines 150 and 151:

Why are values n=6, m=1, and m=0 used? These choices need justifying.

Response:

For these values, we have followed the prescription from reference 10. We have now mentioned it explicitly in the modified text.

The paragraph starting on line 165:

Everything is a bit mixed up in terms of the order. I would suggest streamlining the paper a bit more. Show the data, talk through the two observed dissociation channels and which potential energy states are involved, with the aid of a suitable diagram, and then explain the phenomenon that leads to the observed asymmetry in the angular distribution, together with the fitted model. At the moment, the authors keep jumping around between all of these things and adding little snippets of information each time, and it makes the paper difficult to follow.

Response:

We have reorganised the manuscript now following this suggestion by the reviewer.

Line 180 and 181:

Is the description in these lines of the square of the wavefunction?

Response:

Yes, it is the square of the wavefunction. We have now explicitly mentioned it here in the revised manuscript as follows.

We estimate the initial population of each of the states as a function of the transferred energy **as being** proportional to **the square of the corresponding** part of the ground state vibrational wavefunction in the Franck-Condon region.

Line 188:

On this line, what is referred to that becomes half of the closest energy separation between the two relevant curves?

Response:

Here we refer to the complex matrix element. We have added the relevant text in the revised manuscript as follows.

This coupling matrix element effectively becomes half of the closest energy separation between the two relevant adiabatic potential energy curves.

Line 211:

The mention of higher kinetic energy needs a description of higher than what.

Response:

We have replaced the text 'higher kinetic energy' with '**high kinetic energy**'.

Line 217 to 220:

The description regarding the forward-backward asymmetry reported earlier should be in the introduction.

Response:

We disagree with the reviewer's comment here due to the following reasons:

Although the forward-backward asymmetry has been observed earlier in the ion-pair formation in the previous reports, none of them discusses the quantum interference that we discuss in this manuscript. In fact, Van Brunt [14] and Nag and Nandi [16] attribute it to measurement/instrument artefacts. Moreover, arriving at this conclusion from data on many-electron systems like O₂ would not have been possible due to the presence of multiple ion-pair states and a huge number of doubly excited states. Thus the present work is not a follow-up work on O₂ from the literature. In fact, based on our current findings using H₂, we are giving a possible reinterpretation of the previous O₂ work. In view of this, we do not find it suitable to include the O₂ results in the introduction but address them in the conclusion part of the manuscript.

Line 298:

An explanation of why the authors detect H⁻ and not H⁺ be added to the Methods section.

Response:

The following text is added to the earlier part of the 'Results' section instead of the 'Methods' section with two additional references in the revised manuscript:

In principle, the ion pair formation can be studied by observing either the H⁻ or H⁺ signal. However, from 18 eV onwards, H⁺ is also produced by dissociative ionisation, which prevents obtaining a clean signal from the ion pair formation channel [5, 6]. Hence observing H⁻ is the ideal way to study this process.

Minor Comments:

The following corrections have been incorporated.

Line 49:

'environment' is replaced by 'environments'

Line 126:

'...in one go...' is replaced with '...in a single measurement...'

Line 129:

'...the molecular dissociation on electron impact...' is replaced with '...the molecular dissociation products following electron impact...'

Line 131:

'DA process' be replaced with 'dissociative attachment process'.

We have defined the term DA process before this and hence keep the abbreviation as it is.

Line 133:

'...with the electron energy. It is predominantly due to...' is replaced with '...with the electron energy, predominantly due to...'

Line 135:

'...inversion symmetry. It implies that...' is replaced with '*...inversion symmetry. This implies that ...*'

Line: 145

'is' is replaced with '*are*'

Equations 4 and 6:

We have sorted out the font in these equations.

Lines 167 and 168:

'...states would contribute...' is replaced with '*...states contribute...*'

'...overlap with these states would extend...' is replaced with '*...overlap with these states extends...*'

'...making them dissociate...' is replaced with '*...yielding dissociation products...*'

Lines 178, 180, 183, and 185:

'...resonant process and would occur at electron energies above its threshold. In this model, we consider...' is replaced with '*...resonant process and occurs at any electron energy above its threshold. In the following model, we consider...*'

'...the transferred energy proportional to ...' is replaced with '*...the transferred energy as being proportional to...*'

'...each state that would survive autoionisation. Subsequently, this wavepacket would dissociate along ...' is replaced with '*...each state that survives autoionisation. Subsequently, this wavepacket dissociates along...*'

'...multiplying the survived wavepacket...' is replaced with '*...multiplying the surviving wavepacket...*'

Line 201, 202, and 210:

'...by adding the integrated value of $I(\vartheta)$ from equation (6) over 0 to $\pi/2$ and $\pi/2$ to π , respectively, for each value of K_f .' is replaced with '*...by integrating $I(\vartheta)$ from equation (6) over the range 0 to $\pi/2$ and $\pi/2$ to π , respectively, and summing over the values of K_f .*'

'The heavier isotope would affect the process in terms of the reduced amplitudes of the interfering wavepackets and the relative phase between them.' is replaced with '*The heavier isotope yields reduced amplitudes of the interfering wavepackets and alter the relative phase between them.*'

Apart from the changes mentioned above, we have rearranged the manuscript based on some of the suggestions from reviewer 2. We have also tweaked the language in a few places. All the changes in the text of the manuscript are kept in red colour fonts. Additionally, we have also changed the order of references due to the changes in the structure of the manuscript. These changes, along with their reference numbers, are incorporated in the revised manuscript.

Reviewer #1 (Remarks to the Author):

The authors addressed my question and I believe the modifications, also in response to the comments of the other referee, have improved the manuscript.

In the new version of the manuscript, there are now two panels in Figure 1 to explain the low and high kinetic energy H⁻ signals. This is helpful, but at first glance this figure is confusing: The potentials are labeled with the point-group symmetry labels only, so it may not be immediately clear that each panel is showing only the subset of potentials needed to explain a specific mechanism, and it may even give the impression that the potentials somehow depend on the kinetic energy of the ion, which would be very confusing. Only after comparing the shapes of the potentials to those in Ref 4 does it become clear exactly which electronic states are involved.

I suggest giving the full labels of the states, in particular for the singlet states for $R < 6$ angstrom. It could also help to mention more explicitly in the caption that the different gerade and ungerade singlet states involved in the excitation in the Franck-Condon region lead to different kinetic energies.

Three minor issues:

In Eq. (6), the third spherical harmonic, the p-wave, is written as $Y_{l,0}$. I assume $l=1$ was meant, i.e., $Y_{1,0}$. This comment also applies to Eq. (8).

I furthermore wonder why in these equations the phase of the d-wave is denoted by ϕ_1 , while the p-wave is denoted by ϕ_2 : I would have expected the reverse.

The abbreviation for dissociative attachment (DA) is given on page 12 (line 326), but used already on page 6 (line 150).

Reviewer #2 (Remarks to the Author):

I am content with the changes to the manuscript.

Response to the review reports

We thank the reviewers for their comments and suggestions. Our response to their comments is given below.

Reviewer 1:

Comment:

The authors addressed my question, and I believe the modifications, also in response to the comments of the other referee, have improved the manuscript.

In the new version of the manuscript, there are now two panels in Figure 1 to explain the low and high kinetic energy H- signals. This is helpful, but at first glance this figure is confusing: The potentials are labeled with the point-group symmetry labels only, so it may not be immediately clear that each panel is showing only the subset of potentials needed to explain a specific mechanism, and it may even give the impression that the potentials somehow depend on the kinetic energy of the ion, which would be very confusing. Only after comparing the shapes of the potentials to those in Ref 4 does it become clear exactly which electronic states are involved.

I suggest giving the full labels of the states, in particular for the singlet states for $R < 6$ angstrom. It could also help to mention more explicitly in the caption that the different gerade and ungerade singlet states involved in the excitation in the Franck-Condon region lead to different kinetic energies.

Response:

We have now provided the labels to the curves, and have modified the figure caption accordingly.

Comment:

Three minor issues:

In Eq. (6), the third spherical harmonic, the p-wave, is written as $Y_{l,0}$. I assume $l=1$ was meant, i.e., $Y_{1,0}$. This comment also applies to Eq. (8).

Response:

The equations are corrected.

Comment:

I furthermore wonder why in these equations the phase of the d-wave is denoted by ϕ_1 , while the p-wave is denoted by ϕ_2 : I would have expected the reverse.

Response:

The changes are made in the equations as well as in the text where these parameters are introduced.

Comment:

The abbreviation for dissociative attachment (DA) is given on page 12 (line 326), but used already on page 6 (line 150).

Response:

The term 'dissociative attachment (DA)' is now explicitly mentioned on page 6, where it comes for the first time. Subsequently, the abbreviation 'DA' is used.

Reviewer 2:

Comment:

I am content with the changes to the manuscript.

Response:

We thank the reviewer for the report.